# Cross-Cultural Adaptation and Validation of the Oral Health Values Scale for the Portuguese Population

**DOI:** 10.3390/jpm12050672

**Published:** 2022-04-22

**Authors:** Vanessa Machado, André Mendonça, Luís Proença, José João Mendes, João Botelho, Daniel W. McNeill, Ana Sintra Delgado

**Affiliations:** 1Clinical Research Unit (CRU), Centro de Investigação Interdisciplinar Egas Moniz (CiiEM), Egas Moniz—Cooperativa de Ensino Superior, 2829-511 Monte da Caparica, Portugal; andremendonca@outlook.pt (A.M.); jmendes@egasmoniz.edu.pt (J.J.M.); jbotelho@egasmoniz.edu.pt (J.B.); anasintradelgado@gmail.com (A.S.D.); 2Evidence-Based Hub, CiiEM, Egas Moniz—Cooperativa de Ensino Superior, 2829-511 Monte da Caparica, Portugal; lproenca@egasmoniz.edu.pt; 3Quantitative Methods for Health Research (MQIS), CiiEM, Egas Moniz—Cooperativa de Ensino Superior, 2829-511 Monte da Caparica, Portugal; 4West Virginia University, Morgantown, WV 26506, USA; dmcneil@wvu.edu; 5Orthodontic Department, Egas Moniz Dental Clinic (EMDC), Egas Moniz—Cooperativa de Ensino Superior, 2829-511 Monte da Caparica, Portugal

**Keywords:** oral health, values, psychometric properties

## Abstract

Background: To adapt and validate cross-culturally the Oral Health Values (OHVS) questionnaire to Portuguese language. Methods: The OHVS questionnaire was culturally translated and adapted according to international guidelines. We enrolled 280 patients in a population-based epidemiological survey conducted at the Egas Moniz Dental Clinic (Almada, Portugal). The participants answered the Portuguese version of the OHVS (OHVS-PT), which is a 12-item scale with four-factor structure (Professional Dental Care, Appearance and Health, Flossing and Retaining Natural Teeth factor). Psychometric properties were tested using content validity, construct validity, internal consistency, and test–retest reliability. Results: The OHVS-PT presented adequate reliability (ICC = 0.93, 95% confidence interval (CI): 0.86; 0.97, *p* < 0.001) with values for the Cronbach’s alpha coefficient of the sub-constructs ranging from 0.92 to 0.98. In the Confirmatory Factor Analysis, the final models presented good fit, with the Comparative Fit Indices ranging from 0.882 to 0.891 and the root-mean-squared error of Approximation between 0.065 and 0.069. Conclusions: The OHVS-PT was shown to be a valid and reliable tool to assess oral health values in a Portuguese population. Further studies should evaluate the psychometric properties of the oral personal representation on dental specialties and its impact on dental appointments and procedures.

## 1. Introduction

The value towards health represents the personal view that each one of us places on health and ultimately influences our priority to engage in particular health behaviors or negligent attitudes [1]. This is particularly relevant in oral health because of the multiple oral health care systems in developing and developed countries. In countries where oral health is integrated in the national health systems, hypothetically this may contribute to a greater value attributed to oral health, whereas in countries where oral health is based on insurance-based systems, the apprehend value may differ. Preventive and tooth-preserving strategies are the gold standard public measures [2], yet patients’ adherence to these strategies strongly depend on how they value such care [3,4,5].

In the Portuguese scenario, oral health care is mainly based on a out-of-pocket health system [6], with limited national oral programs for children, adolescents and vulnerable groups (pregnant women, patients with the Human Immunodeficiency Virus and elderly people with low socioeconomic status). This lack of dental services in public hospitals and health centers of the National Health Service [7] may contribute to the high prevalence of oral diseases and support the need for a comprehensive program to assess the personal representation of oral health.

Regarding Oral Health Values (OHV), this is defined as one’s prioritization of or dedication to maintaining or enhancing the particular oral component (gingiva, teeth and orofacial functioning) [8]. Remarkably, the adherence to dental care and appointments is strongly dependent on established OHV [3,4,5], and may have wider repercussions in other health behaviors, such as brushing or interdental cleaning habits, smoking habits or even diet [8]. 

The OHV Scale (OHVS) is a recently developed instrument to measure the value that a person places on his or her own oral health [8]. The psychometric properties of this short tool were attested and correlated with other validated questionnaires such as Oral Health Impact Profile 14 (OHIP-14) [9], the Comprehensive Measure of Oral Health Knowledge (CMOHK) [10], and the Importance of Dental Behaviours (IDB) [11], among others. In total, OHVS comprises four subscales: professional dental care, appearance, flossing, and retention of natural teeth. Its design and content thus present potential in epidemiological and behavioral research in oral health.

Considering OHVS is new and requires geographic expansion to allow future longitudinal analyses and comparability across countries, cross-culturally validating to Portuguese will allow the evaluation of the Portuguese population’s view on oral health-related values and habits. Thus, this study aimed to cross-culturally adapt and validate OHVS to Portuguese (from Portugal), which was nominated OHVS-PT. Herein, we demonstrate that OHVS-PT has content reliability, internal consistency and construct validity.

## 2. Materials and Methods

### 2.1. Design and Participants

The target population of the present cross-sectional study consisted of adults over 18 year of age, Portuguese speakers attending Egas Moniz Dental Clinic (EMDC), a university dental clinic located in Almada, Portugal. Participants were included through a simple random sampling design protocol. This study was conducted in accordance with the Declaration of Helsinki of 1975, as revised in 2013, and received approval by the Institutional Review Board (Ethics Committee of Egas Moniz, ID: 1050). Participants who met these inclusion criteria were invited to participate voluntary and anonymously. Written and informed consent was obtained from each participant before proceeding with the study. The interviewer (A.M.) was blinded to the detailed oral health status. 

### 2.2. Cross-Cultural Adaptation of OHVS Questionnaire

The original OHVS questionnaire measures the value placed by an individual on oral health. This 12-item tool, framed within four-subscales, assesses relevant OHV domains: professional dental care (items 4, 8 and 11); appearance and health (items 3, 7 and 12); flossing (item 2, 5 and 10); and retaining natural teeth (item 1, 6 and 9). Each item is rated using a 5-point scale as follows: 1 = “Strongly disagree”, 2 = “Disagree”, 3 = “Neutral”, 4 = “Agree” and 5 = “Strongly agree” (Table 1). The total score was calculated by summing up the scores for OHVS items with a reverse scoring of items 2, 4, 6, 8, 9, and 11, based on recommendations for scale construction [8]. 

For the purpose of cross-cultural adaptation and validation to Portuguese, an expert panel comprising four independent bilingual individuals fluent in Portuguese and English (including 2 women and 2 men; V.M., J.B., A.D., and A.M) from various oral health backgrounds (general dental practitioner, orthodontist, periodontologists), with years of experience ranging from 2 to 22. Firstly, the original English questionnaire was translated to Portuguese by two native speakers in Portuguese and English (V.M. and J.B.), independently, in a ‘double-blinded’ approach, and were integrated into a single translation version. Any disagreements were solved by discussion. Secondly, two independent bilingual experts, blinded to the original version, back-translated the Portuguese versions. The two new English versions of OHVS were presented to a panel of experts, who assessed inconsistency between the translated versions. The synthesis of the Portuguese version of OHVS tool was found compatible with the original English version in semantic and holistic terms and named as OHVS-PT (Table 1). 

A pilot study was conducted to test the translated OHVS-PT, and to collect feedback from participants. A random sample of 28 individuals (10% of the total sample required for validation, see Section 2.4), who fulfilled the inclusion criteria (living in Portugal, native in Portuguese and 18 years old or older) and consent to participate in the study were included. Because no changes were made, this sample of participants were invited to answer the same test, one week later, for retesting purposes. Additionally, this group of patients did not account for the validation per se. Each participant was provided with the finalized version of OHVS-PT. The OHVS-PT did not require any adjustment based on the feedback, and the participants were recalled after one week for reliability analysis (see Section 2.5.1). 

### 2.3. Sociodemographic Variables

Sociodemographic characteristics comprised age, sex, educational level (elementary, middle or higher), occupation status (student, employed, unemployed or retired), marital status (single, married/union of fact, divorced or widowed) and dental and smoking habits were recorded.

### 2.4. Sample Size Calculation

The sample size was defined according to Terwee et al., ensuring a minimum of 10 individuals per questionnaire item [12]. The total number of subjects (*n* = 280) was determined, taking into account the number of parameters and dimensions present in the questionnaire, in order to ensure an adequate stability of the variance/covariance matrix, when performing a Confirmatory Factor Analysis (CFA). 

### 2.5. Statistical Analyses

#### 2.5.1. Reliability

OHVS-PT reliability analysis was conducted through test–retest reliability and internal consistency analysis using 28 participants (10% of the sample size) who filled the OHVS questionnaire twice with a 1-week interval [12]. The internal consistency was evaluated by calculating Cronbach’s alpha (α) coefficient in R version 1.1-1 (R Studio Team 2018) ‘ltm’ package. An α coefficient of 0.70 was acceptable for the items in the OHVS-PT [13]. The test–retest reliability was calculated with intraclass correlation coefficient (ICC) obtained by the two measurement scores from the participants in R version 0.84.1 (R Studio Team 2018) ‘irr’ package. ICC values were interpreted as follows: excellent (over 0.9), acceptable (over 0.8), weak (over 0.6) and inexistent (below 0.6) [14]. 

#### 2.5.2. Descriptive Analysis and Construct Validity

Descriptive analyses of background characteristics of the target participants, and OHVS items and subscales are presented as counts and correspondent percentages (%), mean and standard deviation (SD), median and interquartile range (IQR), or minimum and maximum values. R version 1.0.8 (R Studio Team 2018) ‘dplyr’ package was used for the descriptive statistical analyses of entire data. The Chi-square and Mann–Whitney tests were used to evaluate the differences in the total sum scores between different sub-groups. The level of statistical significance was set at 5% in all analyses.

CFA was calculated in R version 0.6-10 (R Studio Team 2018) using the ‘lavaan‘ package, to obtain the factorial loads and the model fit of each sub-construct. The maximum likelihood method was applied to calculate the model, and Chi-square (χ2) was used to assess the differences between models, utilizing a likelihood ratio test. Several model fit indices were used to assess the CFA model fit, including the χ2/df ratio (good adjustment with values <2) [15], the root-mean-squared error of approximation (RMSEA; good model adjustment considered for values between 0.05 and 0.10%, 90% confidence interval [CI]) [16], the confirmatory fit index (CFI) (cut-off criterion of ≥0.90 indicates a good fit) [17] and goodness-of-fit (GFI) statistics (values of 0.95 or greater indicate well-fitting models) [18]. 

Then, the invariance of OHVS-PT was explored across sex. We estimated four successive models: (1) unconstrained; (2) factor loadings constrained (Model 1); (3) factor loadings and structural covariances constrained (Model 2); and (4) factor loadings, structural covariances and measurement residuals constrained (Model 3). To measure the invariance, we used the CFI delta values (ΔCFI), with a cut-off point less than 0.01, which indicated invariance [17,19]. The Chi-square delta values (Δχ2) were also used and a value lower than standardized Δχ2 for 1 − α = 0.095 indicated the invariance between the models [20,21]. We also explored the relationships between OHVS-PT items using the Spearman’s rank correlation coefficient.

## 3. Results

### 3.1. Reliability of OHVS

Over a one week interval, all 28 individuals completed the OHVS-PT questionnaire twice, with a 1-week interval. Of these 28 participants, 16 (57.1%) were females and 12 (42.9%) were males, with similar age intervals (females: 39.1 ± 16.7 vs. males: 41.2 ± 16.1, *p* = 0.76), education background, marital status and employment status. The median total score of the OHVS-PT questionnaire was 50 (range: 47–55).

The internal consistency was tested by Cronbach α, with an overall result of 0.97 (95% CI: 0.92; 0.99) (Table 2). Additionally, all subscales demonstrated an acceptable coefficient. The reliability was tested by ICC with a result of 0.93 (95% CI: 0.86; 0.97) (*p* < 0.001). ICC of subscales were all over 0.8, between 0.85 and 0.95. Nominally, two of four subscales had excellent reliability (Flossing subscale = 0.95; Retaining natural teeth = 0.92), and the remaining subscales had acceptable reliability (Professional dental care = 0.85; Appearance and health = 0.89) (Table 2 and Appendix A).

### 3.2. Participant’s Description

A total of 280 participants were included in the study, with an average age of 51.5 (±18.5), mostly married (49.3%, *n* = 138), employed (55.0%, *n* = 154) and non-smoker (42.9%, *n* = 123) (Table 3). The group was predominantly composed of women (53.6%, *n* = 150), yet the sociodemographics did not vary significantly according to sex for mean age (*p* = 0.864), age intervals (*p* = 0.961), education level (*p* = 0.270), marital status (*p* = 0.074), professional activity (*p* = 0.892) and toothbrushing habits in the past 7 days (*p* = 0.069). Only the smoking habits (*p* < 0.001) and interproximal hygiene in the past 7 days (*p* < 0.001) significantly differed between men and women.

Analyzing the results of OHVS-PT, items 1 and 3 had the highest average score, 4.7 (±0.7 and ±0.6, respectively), while items 9 (2.8 ± 1.4) and 10 (3.0 ± 1.0) had the lowest scores. Regarding the subscales, the ‘Appearance and Health’ presented the highest score, 13.9 (±1.6), and ‘Flossing’ the lowest, 9.6 (±3.0) (Table 4).

### 3.3. Construct Validity

#### 3.3.1. Factor Validity

The CFA confirmed the OHVS unifactorial structure (Table 4). The first-order unifactorial model using CFA resulted in an adequate model fit: χ2/df = 1.57, GFI = 0.996, CFI = 0.889, RMSEA = 0.065, CI 90% (0.046–0.082) (Table 5).

#### 3.3.2. Psychometric Analysis

This OHVS-PT questionnaire had an overall adequate reliability (with a Cronbach’s α coefficient = 0.75), thus adequate psychometric properties. Given the unifactorial nature of the OHVS, we were not able to compute the convergent and discriminant validities.

#### 3.3.3. Gender Invariance Measurement

The multigroup CFA was used to explore invariance across gender in OHVS (Table 5). Overall, we concluded the existing of invariance for gender groups for the following reasons: (i) M1, compared to the unconstrained model, had ΔCFI = 0.009 and Δχ2 = 13.101 lower than standardized Δχ2; (ii) M2, compared to the unconstrained model, had ΔCFI = 0.002 and Δχ2 = 17.214 lower than standardized Δχ2; (iii) M2, compared to the unconstrained model, had ΔCFI = 0.002 and Δχ2 = 17.214 lower than standardized Δχ2.

#### 3.3.4. Relationships between OHVS Components

We then assessed whether the items of the OHVS were correlated, through Spearman’s coefficient. We verified a substantial high number of significant correlations (86.4% of the correlations, 57 out of 66) (Table 6). We further assessed the correlation between the subscales, confirming significant correlations among all subscales (Table 7).

## 4. Discussion

The results of this validation study show that the OHVS-PT provided patients’ attribute value to oral health. Despite its short dimension, OHVS-PT showed adequate internal consistency and reliability. The total score and four-subscale scores of OHVS-PT appear to have reliability and validity to measure OHV. 

The validation of this questionnaire may gain more relevance in the Portuguese scenario because several studies have reported credible signs of negligent behaviors towards oral health. On the one hand, levels of negligence towards periodontal diagnosis despite the report of possible signs of periodontitis were reported around 69% [22]. On the other hand, the self-perception of periodontal condition even after receiving the diagnosis of periodontitis was observed to be very poor, and this is detrimental for the initial periodontal treatment adherence [23,24]. Therefore, the OHVS may strengthen the holistic understanding of the psychological and social/environmental factors within oral health beliefs [25], and enable the development of future intervention behavioral oral health strategies.

How the OHVS might be applied and what the potential impacts are is an important issue to address. In our view, this tool may be employed in a two-stage process. First, the initial interaction with OHVS (that is, the first time a person answers this questionnaire) may provide a snapshot on the personal representation one places on oral health. At this stage, multiple studies on different dental specialties are anticipated to be performed in the future. Second, subsequent interactions with the OHVS will depict the impact of dental appointments and procedures on this personal view. With this in mind, one might argue whether behavioral interventions could result in positive improvements on OHV, and this is a matter of further research.

### Strengths and Limitations

Concerning strengths, the relatively short extension of OHVS-PT may contribute to considerable response rates [26], thus contributing to a high interest being applied in the daily clinical practice. The studied population was interviewed before the delivery of any treatment or diagnosis procedure, and the retest interview window timeline was set at an appropriate time [27]. Additionally, patients were interviewed in several specialty appointments, thus increasing the diversity of the validation sample. Furthermore, the mode of questionnaire administration was face-to-face interview, increasing population coverage, reducing the cognitive burden, high response and questionnaire completion, higher willingness to disclose sensitive information [28]. However, there are important potential biases to highlight, for instance, higher social desirability bias, “yes-saying” bias and interviewer bias [28]. To minimize them, the questionnaire was delivered to patients in privacy and without interference. Furthermore, these results should be interpreted with caution given the lack of national representativeness.

In addition, this study of validation was held in a single dental clinic localized at the Southern Lisbon Metropolitan Area; however the cross-cultural validation has, in our view, an almost minimal degree of bias, due to the relative cultural and linguistic homogeneity in Portugal.

## 5. Conclusions

The OHVS-PT showed adequate reliability on repeated questionnaire answers and internal consistency. Considering the short extent and ease of applicability, these findings are relevant for both the academic and clinical contexts.

## Figures and Tables

**Table 1 jpm-12-00672-t001:** Original and Portuguese versions of the OHVS questionnaire.

	Original	Portuguese
Item 1	It is important to me to keep my natural teeth.	É importante para mim manter os meus dentes naturais.
Item 2	It is okay for me to miss a day or two of flossing when I am busy. ^R^	Não há problema se não usar fio dentário durante um ou dois dias quando estou ocupado(a).
Item 3	My smile is an important part of my appearance.	O meu sorriso é uma parte importante da minha aparência.
Item 4	Going to a dentist is not worth the cost to me. ^R^	Ir ao dentista não vale o esforço financeiro que é necessário. ^R^
Item 5	Flossing my teeth every day is a high priority for me.	Usar fio dentário todos os dias é uma grande prioridade para mim.
Item 6	I would rather get dentures than spend money to treat cavities or gum disease. ^R^	Prefiro ter uma prótese dentária a gastar dinheiro a tratar cáries ou doença gengival. ^R^
Item 7	I think it is important that my teeth and gums are a source of pride.	Acredito que é importante que os meus dentes e gengivas sejam um motivo de orgulho.
Item 8	If I have a toothache, I prefer to wait and see if it will go away on its own before seeing a dentist. ^R^	Se tenho dores de dentes, prefiro esperar que a dor passe antes de ir ao dentista. ^R^
Item 9	I would not mind if I had to have a false tooth or dentures. ^R^	Não me importaria de ter um dente falso ou uma prótese dentária. ^R^
Item 10	I make sure I have dental floss available with me so I have it when I need it.	Certifico-me de que tenho o fio dentário comigo para usá-lo quando precisar.
Item 11	Going to the dentist is only important if my teeth or gums are bothering me. ^R^	Ir ao dentista só é importante se os meus dentes e gengivas me estão a incomodar. ^R^
Item 12	The condition of my teeth and gums is an important part of my overall health.	A situação dos meus dentes e gengivas é uma parte importante da minha saúde em geral.

Note: ^R^ Indicates items that are reverse scored. These twelve items are rated on a response scale: 1 = “Strongly disagree”, 2 = “Disagree”, 3 = “Neutral”, 4 = “Agree”, and 5 = “Strongly agree” (in Portuguese: 1 = “Discordo totalmente”; 2 = “Discordo”; 3 = “Não concordo nem discordo”; 4 = “Concordo”; 5 = “Concordo totalmente”).

**Table 2 jpm-12-00672-t002:** Test–retest reliability using ICCs for the OHVS-PT questionnaire.

	Cronbach’s α Coefficient (95% CI)	ICC (95% CI)	*p*-Value
Sub-scales	0.92 (0.75; 0.99)	0.85 (0.71; 0.93)	<0.001
Professional dental care	0.94 (0.80; 0.98)	0.89 (0.77; 0.95)	<0.001
Appearance and health	0.98 (0.94; 0.99)	0.95 (0.90; 0.98)	<0.001
Flossing	0.96 (0.90; 0.99)	0.92 (0.84; 0.96)	<0.001
Retaining natural teeth	0.97 (0.92; 0.99)	0.93 (0.86; 0.97)	<0.001
Total score	0.92 (0.75; 0.99)	0.85 (0.71; 0.93)	<0.001

Abbreviations: CI—confidence interval; ICC—intraclass correlation coefficient.

**Table 3 jpm-12-00672-t003:** Sociodemographic characteristics of the included participants (*n* = 280).

	Total	Female (*n* = 150)	Male (*n* = 130)	*p*-Value
Age, mean (SD)	51.5 (18.5)	51.8 (18.5)	51.2 (19.3)	0.864 *
Age interval, *n* (%)				
18–30	54 (19.3)	28 (18.7)	26 (20.0)	0.961 ^#^
31–40	31 (11.1)	15 (10.0)	16 (12.3)	
41–50	32 (11.4)	18 (12.0)	14 (10.8)	
51–60	65 (23.2)	36 (24.0)	29 (22.3)	
61–70	52 (18.6)	30 (20.0)	22 (16.9)	
71–80	36 (12.9)	18 (12.0)	18 (13.8)	
>80	10 (3.6)	15 (3.3)	15 (3.8)	
Education, *n* (%)				
Elementary	78 (27.9)	36 (24.0)	42 (32.3)	0.270 ^#^
Middle	88 (31.4)	48 (32.0)	40 (30.8)	
Higher	114 (40.7)	66 (44.0)	48 (36.9)	
Marital Status, *n* (%)				
Single	88 (31.4)	47 (31.3)	41 (31.5)	0.074 ^#^
Married	138 (49.3)	72 (48.0)	66 (50.8)	
Divorced	34 (12.1)	15 (10.0)	19 (14.6)	
Widowed	20 (7.1)	16 (10.7)	4 (3.1)	
Professional activity, *n* (%)				
Student	30 (10.7)	18 (12.0)	12 (9.2)	0.892 ^#^
Unemployed	18 (6.4)	10 (6.7)	8 (6.2)	
Employed	154 (55.0)	81 (54.0)	73 (56.2)	
Retired	78 (27.9)	41 (27.3)	37 (28.5)	
Smoking habits, *n* (%)				
Non-smoker	123 (43.9)	88 (58.7)	35 (26.9)	<0.001 ^#^
Former smoker	93 (33.2)	31 (20.7)	62 (47.7)	
Active smoker	64 (22.9)	31 (20.7)	33 (25.4)	
Toothbrushing last 7 days, mean (SD)	6.8 (0.9)	6.8 (0.6)	6.7 (1.3)	0.069 *
Interproximal hygiene last 7 days, mean (SD)	3.6 (2.9)	4.2 (2.8)	2.9 (2.8)	<0.001 *

Abbreviations: OHVS-PT—Oral Health Value Scale-Portuguese; SD—standard deviation. Data is presented as mean values ± SD for continuous variables, and as frequency counts (%) for categorical variables. * Mann–Whitney U test, ^#^ Chi-square test.

**Table 4 jpm-12-00672-t004:** Descriptive statistics of OHVS scores (mean, standard deviation (SD), median and interquartile range (IQR), minimum and maximum).

	Mean (SD)	Median (IQR)	Min–Max
OHVS-PT Total Score	47.2 (6.8)	48.0 (10.3)	31–60
Professional Dental Care subscale	12.2 (2.6)	13 (3)	3–15
Item 4	4.3 (1.1)	5 (1)	1–5
Item 8	4.2 (1.1)	5 (1)	1–5
Item 11	3.8 (1.3)	4 (2)	1–5
Appearance and Health subscale	13.9 (1.6)	15 (2)	7–15
Item 3	4.7 (0.6)	5 (0)	1–5
Item 7	4.5 (0.9)	5 (1)	1–5
Item 12	4.6 (0.7)	5 (1)	1–5
Flossing subscale	9.6 (3.0)	10 (5)	3–15
Item 2	3.2 (1.2)	3 (2)	1–5
Item 5	3.4 (1.3)	4 (1)	1–5
Item 10	3.0 (1.3)	3 (2)	1–5
Retaining Natural Teeth subscale	11.6 (2.4)	12 (3)	3–15
Item 1	4.7 (0.7)	5 (1)	1–5
Item 6	4.1 (1.3)	5 (1)	1–5
Item 9	2.8 (1.4)	3 (2)	1–5

Abbreviations: IQR—interquartile range; OHVS-PT—Oral Health Value Scale-Portuguese; SD—standard deviation.

**Table 5 jpm-12-00672-t005:** Model fit indices in the unifactorial model and configurational invariance by sex.

Description	χ2	df	χ2/df	CFI	GFI	RMSEA (90% CI)	ΔCFI	Δχ2	df
Unifactorial model	176.357 *	112	1.57	0.889	0.996	0.065 (0.046–0.082)	-	-	-
Measurement invariance across sex									
Unconstrained	159.143 *	96	1.66	0.891	0.997	0.069 (0.049–0.088)	-	-	-
Model 1	172.244 *	104	1.66	0.882	0.996	0.069 (0.050–0.087)	0.009	13.101	8
Model 2	176.357 *	112	1.57	0.889	0.996	0.065 (0.046–0.082)	0.002	17.214	8
Model 3	176.357 *	112	1.57	0.889	0.996	0.065 (0.046–0.082)	0.002	17.214	8

Abbreviations: CFI, confirmatory fit index; CI, confidence interval; df, degrees of freedom; GFI, goodness of fit index; RMSEA, root-mean-square error of approximation; χ2, Chi-square. Model 1, factor loadings constrained; Model 2, factor loadings and structural covariances constrained; Model 3, factor loadings, structural covariances and measurement residuals constrained. * *p* < 0.01.

**Table 6 jpm-12-00672-t006:** Correlation between OHVS item scores.

Items	1	2	3	4	5	6	7	8	9	10	11	12
**1**	1.000	0.043	0.120 *	0.095	0.190 **	0.178 **	0.197 ***	0.149 *	0.050	0.189 **	0.134 *	0.109
**2**	-	1.000	0.248 ***	0.231 ***	0.384 ***	0.284 ***	0.191 **	0.202 **	0.159 **	0.306 ***	0.219 ***	0.205 ***
**3**	-	-	1.000	0.156 **	0.272 ***	0.186 **	0.421 ***	0.132 *	0.035	0.099	0.237 ***	0.398 ***
**4**	-	-	-	1.000	0.278 ***	0.383 ***	0.173 **	0.229 ***	0.164 **	0.273 ***	0.366 ***	0.192 **
**5**	-	-	-	-	1.000	0.265 ***	0.313 ***	0.322 ***	0.151 *	0.652 ***	0.227 ***	0.252 ***
**6**	-	-	-	-	-	1.000	0.245 ***	0.312 ***	0.384 ***	0.172 **	0.253 ***	0.088
**7**	-	-	-	-	-	-	1.000	0.240 ***	0.111	0.224 ***	0.298 ***	0.356 ***
**8**	-	-	-	-	-	-	-	1.000	0.184 **	0.264 ***	0.328 ***	0.145 *
**9**	-	-	-	-	-	-	-	-	1.000	0.194 **	0.213 ***	−0.024
**10**	-	-	-	-	-	-	-	-	-	1.000	0.254 ***	0.199 ***
**11**	-	-	-	-	-	-	-	-	-	-	1.000	0.302 ***
**12**	-	-	-	-	-	-	-	-	-	-	-	1.000

Values are the Spearman’s rank correlation coefficient (rho), * *p* < 0.05 (colored as yellow). ** *p* < 0.01 (colored as orange). *** *p* < 0.001 (colored as green).

**Table 7 jpm-12-00672-t007:** Correlation between OHVS subscales scores.

OHVS	Professional Dental Care	Appearance and Health	Flossing	Retaining Natural Teeth
Professional Dental Care	1.000	0.402 ***	0.440 ***	0.425 ***
Appearance and Health	-	1.000	0.391 ***	0.201 ***
Flossing	-	-	1.000	0.331 ***
Retaining Natural Teeth	-	-	-	1.000

Values are the Spearman’s rank correlation coefficient (rho), *** *p* < 0.001.

## Data Availability

Not applicable.

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
