# Peer review of "Cross-Cultural Adaptation and Validation of the Oral Health Values Scale for the Portuguese Population"

_jpm, 2022, doi:10.3390/jpm12050672_

Round 1

Reviewer 1 Report

Introduction
Try to emphasize the importance of the use of this scale in dental services and to explain why it is relevant specifically in Portugal, providing data from the country to understand the context.
Material and methods
What were the exclusion criteria? Please specify in the manuscript.
Was the clinic where the study was conducted private or public, and could this have biased the results?
Give the methods of selection of participants
Discussion
The results obtained in your study should be compared with similar studies. What other languages have you translated in? What results have you obtained? Are they similar? How do they differ? What could be the reasons for these differences?
In line 245... "The validation of this questionnaire may gain more relevance in the Portuguese scenario because several studies have reported credible signs of negligent behaviors towards oral health". Why is it more relevant in the Portuguese scenario?
To indicate in the limitations section that it would have been interesting and would have increased the value of the study if medical markers of oral health (dmf, gingival index, malocclusions, bruxism, etc) had been measured. 
In addition, it should be mentioned that this is not a representative sample and the results should be interpreted with caution.

Author Response

Dear Prof. Dr. David Alan Rizzieri

Editor-in-Chief of Journal of Personalized Medicine

We are pleased with the opportunity to revise and resubmit our manuscript entitled Cross-cultural adaptation and validation of the Oral Health Values Scale for the Portuguese population (jpm-1684665).

Considering the editor and reviewers’ comments, all have been considered very important and were thoroughly considered.

We hope the revised manuscript will enable its further consideration. We are happy to consider further revisions and we thank you for your continued interest in our research.

Reviewer 1:

Introduction

Try to emphasize the importance of the use of this scale in dental services and to explain why it is relevant specifically in Portugal, providing data from the country to understand the context.

Our response: We strongly agree with the reviewer that it is important to explain  the Portuguese panorama regarding the dental health system. We have added information, by saying: “In the Portuguese scenario, oral health care is mainly based on a out-of-pocket health system [6], with limited national oral programs for children, adolescents and vulnerable groups (pregnant women, patients with the Human Immunodeficiency Virus and elderly people with low socioeconomic status). This lack of dental services in public hospitals and health centers of the National Health Service [7] may contribute to the high prevalence of oral diseases and support the need for a comprehensive program to assess the personal representation of oral health.” (line 44-50)

Material and methods

What were the exclusion criteria? Please specify in the manuscript.

Our response: Thank you for this valid question. We opted to describe only the inclusion criteria, by saying: "The target population of the present cross-sectional study consisted of adults over 18 year of age, Portuguese speakers attending Egas Moniz Dental Clinic (EMDC), a private university dental clinic located in Almada, Portugal". We believe that adding exclusion criteria will not add relevant information to the manuscript, as they are the opposite of inclusion criteria.

Was the clinic where the study was conducted private or public, and could this have biased the results?

Our response: We appreciate this useful observation however the clinic is associated with a private university (yet without restrictions of access to the general population), with accessible and affordable costs allowing all population to access the clinic. In Portugal, the dental health system is based on an out-of-pocket model, therefore this could not have biased the results.

Give the methods of selection of participants

Our response: Participants were selected by simple random sampling. We added this information by saying: “Participants were included through a simple random sampling design protocol.” (Line 76-77)

Discussion

The results obtained in your study should be compared with similar studies. What other languages have you translated in? What results have you obtained? Are they similar? How do they differ? What could be the reasons for these differences?

Our response: Thank you for this remark. However, this is the first validation study of a translated version of OHVS for a non-english language. Therefore, such comparison with other results was not possible to achieve.

In line 245... "The validation of this questionnaire may gain more relevance in the Portuguese scenario because several studies have reported credible signs of negligent behaviors towards oral health". Why is it more relevant in the Portuguese scenario?

Our response: As stated in the quoted sentence, this questionnaire may be more relevant because Portuguese patients have been reported by our group’s work with worrisome signs of negligence towards oral health. One possible reason may be the value that these people place in oral health, thus OHVS conveying the personal view on oral health values could be an instrument useful to explore this in future research.

To indicate in the limitations section that it would have been interesting and would have increased the value of the study if medical markers of oral health (dmf, gingival index, malocclusions, bruxism, etc) had been measured. 

Our response: Although adding medical markers of oral health (dmf, gingival index, malocclusions, bruxism, etc) could be interesting, this study focused only on the validation of the OHVS-PT. Only after the successful validation of OHVS-PT, the next steps will be to apply this instrument to a large sample and analyze the association with these markers of oral health in a cross-sectional and cohort design.

In addition, it should be mentioned that this is not a representative sample and the results should be interpreted with caution.

Our response: Thank you for this valid remark. We added this accordingly in the text as follows: “Furthermore, these results should be interpreted with caution given the lack of national representativeness” (line 287-288).

Reviewer 2 Report

The Oral Health Values Scale is useful to measure the value of the person's oral health. Thus, it is important to consider the reliability, internal consistency, and configuration validity of the OHVS-PT content. I think this manuscript is suitable for publication in the present form.

Author Response

Dear Prof. Dr. David Alan Rizzieri

Editor-in-Chief of Journal of Personalized Medicine

We are pleased with the opportunity to revise and resubmit our manuscript entitled Cross-cultural adaptation and validation of the Oral Health Values Scale for the Portuguese population (jpm-1684665).

Considering the editor and reviewers’ comments, all have been considered very important and were thoroughly considered.

We hope the revised manuscript will enable its further consideration. We are happy to consider further revisions and we thank you for your continued interest in our research.

Reviewer 2:

The Oral Health Values Scale is useful to measure the value of the person's oral health. Thus, it is important to consider the reliability, internal consistency, and configuration validity of the OHVS-PT content. I think this manuscript is suitable for publication in the present form.

Our response: Thank you for your encouraging remarks and for your time invested in the revision.

Reviewer 3 Report

Dear authors; 

This is a well conducted study

Hopefully it will help practitioners in Portugal in their future work.

Author Response

Dear Prof. Dr. David Alan Rizzieri

Editor-in-Chief of Journal of Personalized Medicine

We are pleased with the opportunity to revise and resubmit our manuscript entitled Cross-cultural adaptation and validation of the Oral Health Values Scale for the Portuguese population (jpm-1684665).

Considering the editor and reviewers’ comments, all have been considered very important and were thoroughly considered.

We hope the revised manuscript will enable its further consideration. We are happy to consider further revisions and we thank you for your continued interest in our research.

Reviewer 3:

This is a well conducted study.

Hopefully it will help practitioners in Portugal in their future work.

Our response: Thank you for your encouraging remarks and for your time invested in the revision.

Round 2

Reviewer 1 Report

I consider that the authors have correctly made all the suggested changes. Therefore, the manuscript is suitable for publication.

Congratulations!